# Identification of Chlorophyll Metabolism- and Photosynthesis-Related Genes Regulating Green Flower Color in Chrysanthemum by Integrative Transcriptome and Weighted Correlation Network Analyses

**DOI:** 10.3390/genes12030449

**Published:** 2021-03-21

**Authors:** Hansen Fu, Tuo Zeng, Yangyang Zhao, Tingting Luo, Huijie Deng, Chenwei Meng, Jing Luo, Caiyun Wang

**Affiliations:** Key Laboratory for Biology of Horticultural Plants, Ministry of Education, College of Horticulture and Forestry Sciences, Huazhong Agricultural University, Wuhan 430070, China; fhs4242021@163.com (H.F.); tuozeng@webmail.hzau.edu.cn (T.Z.); shane0828@163.com (Y.Z.); luotingting@leoking.com (T.L.); huijiedeng@springwoods.com (H.D.); 15927390348@163.com (C.M.); ljcau@mail.hzau.edu.cn (J.L.)

**Keywords:** florist’s chrysanthemum, green ray floret, segregating population, transcriptome, photosynthesis, chlorophyll metabolism, weighted gene co-expression network analysis (WGCNA)

## Abstract

Green chrysanthemums are difficult to breed but have high commercial value. The molecular basis for the green petal color in chrysanthemum is not fully understood. This was investigated in the present study by RNA sequencing analysis of white and green ray florets collected at three stages of flower development from the F_1_ progeny of the cross between *Chrysanthemum × morifolium* “Lüdingdang” with green-petaled flowers and *Chrysanthemum vistitum* with white-petaled flowers. The chlorophyll content was higher and chloroplast degradation was slower in green pools than in white pools at each developmental stage. Transcriptome analysis revealed that genes that were differentially expressed between the two pools were enriched in pathways related to chlorophyll metabolism and photosynthesis. We identified the transcription factor genes *CmCOLa*, *CmCOLb*, *CmERF*, and *CmbHLH* as regulators of the green flower color in chrysanthemum by differential expression analysis and weighted gene co-expression network analysis. These findings can guide future efforts to improve the color palette of chrysanthemum flowers through genetic engineering.

## 1. Introduction

Florist’s chrysanthemum originated in China and has been cultivated for more than 1500 years [1]. As one of the most popular ornamental flowers in the world, chrysanthemum is sold as a cut flower or cultivated as a potted flower or garden plant, and the flowers are also consumed as a tea. While thousands of chrysanthemum cultivars in a variety of colors are available, green-colored chrysanthemum flowers are relatively rare. Flower color is an important trait that determines the commercial value of chrysanthemum cultivars, and green cultivars are admired for their beauty and rarity.

Flower color is mainly determined by anthocyanin, carotenoid, and chlorophyll contents. The first two are the major pigments in flowers and their synthesis and regulatory mechanisms are well established [2,3,4]. Given the important function of chlorophyll in photosynthesis, chlorophyll metabolism in leaves has been extensively studied with the aim of delaying leaf senescence [5,6]. However, less is known about the role of chlorophyll in green flowers. Ornamental plants with green flowers have been reported; carnations (*Dianthus caryophyllus*) with pale green petals were found to contain more chlorophyll than cultivars with non-green petals [7], and the green color of chrysanthemum florets was also shown to be derived from chlorophyll [8,9].

In chlorophyll metabolism, glutamyl-tRNA is transformed into glutamate-1-semialdehyde by glutamyl-tRNA reductase and then into chlorophyll *a* by glutamate 1-semialdehyde aminotransferase, porphobilinogen synthase, and other enzymes; the transformation of chlorophyll *a* to chlorophyll *b* is carried out by chlorophyll *a* oxygenase, chlorophyll *b* reductase, and hydroxymethyl chlorophyll *a* reductase, and chlorophyll *a* is broken down by chlorophyllase, metal-chelating substance, pheophytinase, pheophorbide *a* oxygenase, and red chlorophyll catabolite reductase [10,11,12]. Changes in the expression of chlorophyll metabolism-related genes were shown to affect the green color of flowers in bicolor lily “Tiny Padhye” (*Lilium* spp.) [13] and the chrysanthemum (*Chrysanthemum morifolium*) cultivars “Feeling White” and “Feeling Green” [14].

Chlorophyll metabolism is affected by plant developmental stage, light and hormone levels, and other factors. Abscisic acid and dark treatment were shown to accelerate chlorophyll degradation, while exogenous application of melatonin reduced the expression of genes associated with chlorophyll catabolism [15]. Moreover, chlorophyll content decreased during the development of plant organs [6,16]. Under heat stress, overexpression of SUPPRESSOR OF OVEREXPRESSION OF CONSTANS 1 (SOC1) and SOC1-like transcription factors (TFs) led to upregulation of chlorophyll biosynthesis-related genes and induced chloroplast biogenesis and the formation of green flower petals in *Nicotiana tabacum* and *Arabidopsis thaliana* [17]. Genes encoding myeloblastosis113 (MYB113), CONSTANS-like 16 (COL16), and ethylene response factor (ERF), whose expression is closely correlated with chlorophyll content, were identified in chrysanthemum [14], but the functions of these genes in the formation of green petals in chrysanthemum have not yet been reported.

The cells of green flower petals contain plastids with a well-developed thylakoid membrane system, while the thylakoid membranes of white florets have been destroyed [14]. Similarly, green disc florets of chrysanthemum have a larger chloroplast and a well-defined grana thylakoid membrane structure compared to white florets during flower development [9]. Thus, the integrity of chloroplasts and the photosynthetic system is closely related to petal color. Identifying the components of chlorophyll metabolism and photosynthesis pathways and clarifying their interactions can provide insight into the molecular mechanisms underlying green flower formation.

Weighted gene co-expression network analysis (WGCNA) is a method for identifying functionally related genes [18] that has been applied to transcriptome analyses in many plants, including rice [19], sweet orange [20], apple [21], and hot pepper [22]. WGCNA has also been used to identify genes that are potentially related to specific traits in ornamental plants—for example, TFs involved in chlorophyll metabolism in lily [13]; genes associated with disease resistance in chrysanthemum [23], and TFs regulating carotenoid accumulation and flower color in *Chrysanthemum* “Jianliuxiang Pink” and its bud sport mutants [24]. However, WGCNA has yet to be applied to the study of green flower color in chrysanthemum.

In this study, we established a segregating F_1_ population of *Chrysanthemum* × *morifolium* “Lüdingdang” crossed with *Chrysanthemum vistitum*, which have green and white flower petals, respectively. Extreme white and green ray florets were collected from F_1_ plants for transcriptome analysis, and WGCNA was carried out to identify genes that were differentially expressed between the two colors. Functional enrichment analysis revealed that the differentially expressed genes (DEGs)—including some encoding TFs—were related to chlorophyll metabolism and photosynthesis pathways.

## 2. Materials and Methods

### 2.1. Plant Materials

A segregating F_1_ population from *Chrysanthemum × morifolium* “Lüdingdang” (green, female) crossed with *C. vistitum* (white, male) was established in 2016 [25], and 537 flowered individuals were selected for experiments. The parental and progeny plants were maintained at the planting sites of Huazhong Agricultural University in Wuhan, China.

The green and white phenotypes were confirmed using a colorimeter (Model CM-5; Konica Minolta, Tokyo, Japan) [25]; ray florets with an *a** value of less than −8.00 were considered as green, and those with an *a** value greater than −2.00 were considered white. We selected 24 green lines for the green pools; at each stage of the flowering period, ray floret samples were obtained from eight lines and pooled together as one biological sample, and three independent biological replicates were collected for each stage for RNA sequencing (RNA-seq) and other analyses. The same sample collection process was used for the white pools. The three stages of flower development were S1 (bud), S2 (vertical ray florets in the outermost whorl), and S3 (horizontal ray florets in the outermost whorl) (Figure 1A). Collected samples were immediately frozen in liquid nitrogen and stored at −80℃ until use. The samples were named according to floret color, flower development stage, and sample number—e.g., WS1-1 (white pool, stage S1, sample 1). Additionally, commercial chrysanthemum cultivars were purchased from a local flower market in Wuhan; three cultivars with white ray florets were named “W-A” (short flat ray florets), “W-B” (long flat ray florets), and “W-C” (short curved ray florets) and three cultivars with green ray florets were named “G-A” (short curved ray florets), “G-B” (long curved ray florets), and “G-C” (short flat ray florets).

### 2.2. Measurement of Chlorophyll Content

Total chlorophyll was extracted from each sample by incubation in 80% acetone in a 60 ℃ water bath for 2 h, followed by spectrophotometric measurement at 663, 645, and 470 nm [26] using an ultraviolet spectrophotometer (Model TU-1810; Beijing Persee General Instrument Co., Beijing, China).

### 2.3. Analysis of Plastid Ultrastructure by Transmission Electron Microscopy

The ultrastructure of chloroplasts in both green and white pools at the three flower development stages was examined by TEM. Tissue samples were obtained from the central part of the petals of the flower head and cut into 1- to 2-mm^3^ pieces that were immersed in 2.5% glutaraldehyde and stored at 4 °C. The samples were fixed, dehydrated, and embedded in Epoxy 821, and ultrathin sections were cut with a diamond knife on an ultramicrotome (Model UC7; Leica Microsystems, Wetzlar, Germany), stained with uranyl acetate and lead citrate, and observed under a TE microscope (G 20 TWIN; FEI Tecnai, Hillsboro, OR, USA) at an accelerating voltage of 200 kV.

### 2.4. RNA Extraction and Sequencing

Total RNA was isolated from the ray florets of the 18 samples using the EASYspin Plant RNA Kit (Aidlab Biotech, Beijing, China), and 1 µg RNA per sample was used as input material for RNA-seq. Sequencing libraries were generated using the NEBNext Ultra RNA Library Prep Kit for Illumina (New England Biolabs, Ipswich, MA, USA) according to the manufacturer’s instructions, and index codes were added to attribute sequences to each sample. Clustering of the index-coded samples was performed on the cBot Cluster Generation System using the TruSeq PE Cluster Kit v3-cBot-HS (both from Illumina, San Diego, CA, USA) according to the manufacturer’s instructions. After cluster generation, library preparations were sequenced on an Illumina Novaseq platform and 150-bp paired-end reads were generated. Single-molecule real-time genome sequencing data from one of the progeny individuals were used for correction.

Reference genome and gene model annotation files of *Chrysanthemum seticuspe* were downloaded from the genome website (https://plantgarden.jp/ja/list/t1111766 accessed on 26 February 2021) [27]. Indexing of the reference genome was performed using Hisat2 v2.0.5, which was also used to align paired-end clean reads to the reference genome [28]. In order to ensure the comprehensiveness of gene annotation, we searched the assembled unigenes against seven public databases including Gene Ontology (GO), Kyoto Encyclopedia of Genes and Genomes (KEGG) Orthology (KO), Eukaryotic Orthologous Groups (KOG), NCBI non-redundant protein sequence (Nr), NCBI nucleotide sequence (Nt), Protein Family (PFAM), and SwissProt protein databases.

Feature Counts v1.5.0-p3 was used to count the number of reads mapped to each gene [29]. The fragments per kilobase of transcript per million mapped reads (FPKM) of each gene were calculated based on the gene length and number of reads mapped to the gene.

### 2.5. Differential Expression Analysis

Differential expression analysis was performed using the DESeq2 R package v1.16.1 [30], which provides statistical algorithms for evaluating differential expression using a negative binomial distribution-based model. The resultant *P* values were adjusted using Benjamini and Hochberg’s approach for controlling the false discovery rate. Genes with an adjusted *P* (P_adj_) <0.01 and |log_2_(fold change [FC])|≥1 were considered as differentially expressed. Heatmaps were generated using the NovoMagic data analysis platform (https://magic.novogene.com/customer/main#/tool-ngs/aa0e7df4dc046c9b6ecfdb73aaf63718/heatmap accessed on 26 February 2021).

### 2.6. TF Identification

TF-encoding unigenes were identified by performing a similarity search against the Plant TF Database (http://plntfdb.bio.uni-potsdam.de/v3.0/downloads.php accessed on 26 February 2021) using BLASTX with an E value cut-off of ≤ 10^−5^ [31].

### 2.7. Weighted Gene Co-Expression Network Analysis

We performed a weighted gene co-expression network analysis (WGCNA) using R package [32] to construct a co-expression network of genes that were differentially expressed between green and white chrysanthemums. A total of 20,949 genes with FPKM > 1 in at least two samples and a coefficient of variation less than 0.5 were screened for WGCNA. After constructing the network, genes with similar expression patterns were classified into modules; those with a topologic overlap metric > 0.3 were considered highly correlated and strongly regulated. Co-expression networks were visualized using Cytoscape_v.3.8.0 software [33].

### 2.8. Quantitative Real-Time-PCR Analysis

To validate the results of RNA-seq, the expression of 13 DEGs related to chlorophyll metabolism including those encoding TFs was analyzed by qRT-PCR. Reverse transcription of RNA samples from the same batch as those used for RNA-seq was performed using the EasyScript One-Step gDNA Removal and cDNA Synthesis Super Mix (TransGen Biotech, Beijing, China); qRT-PCR was performed using the 2× SYBR qPCR mix (Aidlab Biotech, Beijing, China) on a LightCycler 96 Real-Time PCR system (Roche, Basel, Switzerland). The *C. morifolium CmUBI* gene (KF305681) was used as an internal reference, and relative expression levels of target genes were calculated with the 2^−ΔΔCT^ method [34]. Three independent biological replicates of each sample and three technical replicates of each biological replicate were analyzed. Gene-specific primer pairs were designed using Primer Premier v5.0 software (Premier Biosoft, Palo Alto, CA, USA).

## 3. Results

### 3.1. Chlorophyll Content of Green and White Ray Florets of Chrysanthemum

The ray florets in green pools were greener than those in the white pools at each stage of flower development. We assessed the chlorophyll content of the samples at the three stages and found that it was 1.5, 3.5, and 5.5 times higher in the green pools than in the white pools at S1, S2, and S3, respectively. The chlorophyll content of both the green and white florets decreased with the development of flower heads, but the rate of decrease was slower in the green pools than in the white pools (Figure 1B).

### 3.2. Morphology and Structure of Chloroplasts of Green and White Ray Florets

Chloroplasts containing chlorophyll are the site of photosynthesis. We compared the chloroplast morphology of green and white ray florets by TEM to determine whether structural differences accounted for the observed disparity in chlorophyll content. At S1, epidermal and mesophyll cells of both white and green petals contained a chloroplast with intact morphology and a clear thylakoid structure. At S2, the chloroplast in white petals was elongated and became further deformed with the destruction of the thylakoid membrane at S3. In contrast, the structure of chloroplasts in the cells of green petals was normal until S2 but started to change at S3, although the thylakoid membrane remained intact (Figure 2). These results demonstrate that during flower development, chloroplasts are gradually deformed and destroyed—a process that occurs earlier in white as compared to green petals—resulting in different chlorophyll contents.

### 3.3. RNA-Seq Analysis of Green and White Pools

We established 18 sequencing libraries, each with a minimum of 65,518,128 clean reads and at least 9.83 G clean bases after removing reads containing adapter, reads containing ploy-N, and low-quality reads from raw data. Q30 ranged from 93.55% to 94.69% and the GC (guanine-cytosine) content was 41.68–42.45% (Table 1).

Transcript abundance (as estimated by FPKM) among biological replicates was highly correlated, with Pearson’s correlation coefficients between 0.909 and 0.967 (Appendix A). Principal component analysis of the expressed unigenes (FPKM > 0.3) showed that the 18 samples formed three groups according to the flower development stage. The area occupied by S2 was largest, while that occupied by S1 was the smallest (Appendix A). These results indicate that the gene expression profiles of green and white pools were more similar at S1 and showed the greatest difference at S2.

### 3.4. Gene Annotation and Functional Classification

For functional annotation, the assembled unigenes were searched against public databases (GO, KO, KOG, Nr, Nt, PFAM, and SwissProt) using the BLASTx program. A total of 95,783 (54.26%) unigenes were annotated in the GO database along with 22,917 (12.98%) in KO; 13,477 (7.63%) in KOG; 62,514 (35.41%) in Nr; 56,689 (32.11%) in Nt; 95,783 (54.26%) in PFAM, and 69,193 (39.2%) in SwissProt (Appendix A). Additionally, there were 3048 (1.72%) unigenes annotated in all databases and 137,536 (77.91%) in at least one database. The top five annotated species corresponding to unigenes annotated in the Nr database were *Helianthus annuus*, *Lactuca sativa*, *Cynara cardunculus* var. *scolymus*, *Daucus carota* subsp. *Sativus*, and *Cajanus cajan*, accounting for 42.4%, 23.3%, 7.9%, 2.7%, and 2.5% of the total number of Nr annotations, respectively (Appendix A).

### 3.5. DEG Analysis

DEGs with |log_2_FC|≥ 1 and corrected P (P_adj_) < 0.01 were identified from the comparison between green and white pool transcriptomes at the same stage of flower development. In the GS1 vs. WS1 comparison, there were 299 DEGs, including 204 that were upregulated and 95 that were downregulated; in GS2 vs. WS2, there were 2060 DEGs, including 1167 upregulated and 893 downregulated genes; and in GS3 vs. WS3, 1327 DEGs were identified, including 771 that were upregulated and 556 that were downregulated. There were more DEGs at S2 than at S1, but the number decreased at S3; moreover, at each stage, the number of upregulated DEGs was greater than the number of downregulated DEGs. After removing duplicates, there were 2989 DEGs (Figure 3).

We classified the 2989 DEGs according to Gene Ontology Term Enrichment (GO term) and found that the most significant terms were “photosynthesis” in the biological process category; “photosynthesis II oxygen evolving complex” and “thylakoid membrane” in cellular component; and “serine-type peptidase activity” and “serine hydrolase activity” in molecular function (Appendix A). In the KEGG pathway enrichment analysis, the most significant pathways were “Photosynthesis”, “Photosynthesis—antenna proteins”, “Glyoxylate and dicarboxylate metabolism”, and “Porphyrin and chlorophyll metabolism” (Appendix A). Thus, genes associated with photosynthesis and chlorophyll metabolism could account for the different floret colors of the green and white pools. Among the DEGs, 154 encoded TFs in 35 families; six TFs were upregulated in green pools as compared to white pools at each flower development stage. Two of the TFs belonged to the CONSTANS-like (COL) family (Cse_sc021212.1_g020.1 and Cse_sc026878.1_g010.1) and the remaining four were basic helix-loop-helix (bHLH) (Cse_sc007092.1_g020.1), ethylene response factor (ERF) (Cse_sc005439.1_g050.1), transcription activator-like effector (TALE), and MYB family TFs.

### 3.6. Expression of Genes Involved in Chlorophyll Metabolism and Related Pathways

As chlorophyll content directly influences flower color, we screened the unigenes in order to identify those related to chlorophyll metabolism. Of the 113 candidates, 24 were DEGs (Table 2). All of the DEGs related to chlorophyll biosynthesis were more highly expressed in green as compared to white pools (Figure 4); this was confirmed by qRT-PCR analysis, which also showed that the differences in gene expression levels were similar at each stage of flower development (Figure 5).

We identified 650 genes related to photosynthesis (i.e., the “Photosynthesis”, “Photosynthesis—antenna proteins”, “Carbon fixation in photosynthetic organisms”, and “Glyoxylate and dicarboxylate metabolism” KEGG pathways), of which 63 were identified as DEGs (Table 2). All of the DEGs related to photosynthesis were more highly expressed in green pools as compared to white pools (Figure 6), and their expression levels increased during flower development in the former, but not in white pools. These results indicate that the difference in flower color between green and white pools is caused by the differential expression of genes involved in chlorophyll metabolism as well as those related to photosynthesis.

### 3.7. WGCNA of DEGs

From the transcriptome data, we screened 20,949 genes for WGCNA based on FPKM and the coefficient of variation. The genes were divided into the following 12 modules according to similarities in expression patterns (Figure 7A): black (*n* = 587 genes), blue (*n* = 2934), brown (*n* = 2521), green (*n* = 2109), green–yellow (*n* = 243), magenta (*n* = 467), pink (*n* = 479), purple (*n* = 279), red (*n* = 916), tan (*n* = 42), turquoise (*n* = 7997), and yellow (*n* = 2258). Additionally, 117 genes that did not fit into any of these modules were grouped into the gray module, which was omitted from the subsequent analysis. The expression patterns of eigengenes in the 12 modules are shown in Appendix A.

Though the KEGG pathway analysis of all modules, we screened the pathways related to chlorophyll metabolism and photosynthesis in all modules. Genes that were significantly enriched in these pathways were all in the black and red modules (Table 3). The black module showed the highest correlations with phenotype among all of the modules—i.e., a positive correlation with green pools and negative correlation with white pools (Figure 7B). Thus, genes in the black module are likely responsible for the green flower color of chrysanthemum.

### 3.8. Functional Enrichment Analysis of Key Modules

The results of the GO analysis indicated that in the biological process category, the black module was mainly enriched in photosynthesis. In the cellular component category, photosynthesis and related terms were enriched, including photosynthetic membrane, photosystem, thylakoid membrane, photosystem II oxygen evolving complex, oxidoreductase complex, extrinsic component of membrane, and membrane protein complex; these were positively correlated with chlorophyll content (Appendix A).

The KEGG pathway analysis showed that genes in the black module were significantly enriched in six pathways—namely, Photosynthesis, Carbon fixation in photosynthetic organisms, Photosynthesis—antenna proteins, Carbon metabolism, Glyoxylate and dicarboxylate metabolism, and Porphyrin and chlorophyll metabolism (Appendix A). Genes in the red module were enriched in Photosynthesis—antenna proteins.

GO and KEGG pathway analyses were carried out for the DEGs between green and white pools (Appendix A). The results were similar to those obtained for the black module from the WGCNA, confirming the reliability of our data and highlighting the contribution of genes in photosynthesis-related pathways to the regulation of flower color.

In the black module, 27 unigenes were identified as TFs belonging to 19 TF families including ERF (*n* = 4), bHLH (*n* = 2), C2H2 (*n* = 2), COL (*n* = 2), TEOSINTE BRANCHED 1, CYCLOIDEA, PCF1 (TCP) (*n* = 2), Trihelix (*n* = 2), and others (*n* = 13). Six of the TFs were negatively correlated and the others were positively correlated with genes in the key pathways (Appendix A).

### 3.9. Identification of Hub TFs in the Black Module

We selected genes that showed significant enrichment in key pathways and TF genes in the same module to construct a gene regulatory network to identify the hub TFs in the network. In this module, 2741 edges were created. We selected the top 500 edges by their weight value to construct the network, which contained 51 node genes (Figure 8). The network included 6 genes related to Photosynthesis—antenna proteins, 10 related to Carbon metabolism, 5 related to Glyoxylate and dicarboxylate metabolism, 18 related to Photosynthesis, 2 related to Chlorophyll metabolism, and 10 TFs belonging to 7 TF families (ERF, *n* = 3; COL, *n* = 2; C2H2, *n* = 2; APETALA2 (AP2), *n* = 1; M-type MCM1, AGAMOUS, DEFICIENS, and serum response factor (MADS), *n* = 1; basic leucine zipper (b-ZIP), *n* = 1; and NUCLEAR TRANSCRIPTION FACTOR Y SUBUNIT BETA (NFYB), *n* = 1). Two COL (Cse_sc021212.1_g020.1 and Cse_sc026878.1_g010.1) TFs and one ERF (Cse_sc005439.1_g050.1) TF with the highest connectivity were identified as hub TFs in the network.

### 3.10. qRT-PCR Validation of Hub Genes

According to the color of the ray florets of the green pools and the white pools at different stages, we screened 6 TFs by expression analysis and 11 TFs by WGCNA, respectively; three TFs were in common. In addition, in the six TFs screened by differential expression analysis, the TF (Cse_sc007092.1_g020.1) belonging to the bHLH family also showed a similar expression pattern to the phenotype, so we chose these four TFs for further analysis: two were COL family members and were named *CmCOLa* (Cse_sc021212.1_g020.1) and *CmCOLb* (Cse_sc026878.1_g010.1), and the other two were ERF (*CmERF*; Cse_sc005439.1_g050.1) and bHLH (*CmbHLH*; Cse_sc007092.1_g020.1) TFs. The results of the qRT-PCR analysis showed that all of the genes had a higher expression in green as compared to white pools at each flower development stage (Figure 9). We also compared the expression of *CmCOL16a*, *CmCOL16b*, *CmERF*, and *CmbHLH* (Figure 10A) and measured chlorophyll contents (Figure 10B) in three green and three white commercial chrysanthemum cultivars at S3. *CmCOL16b* and *CmbHLH* were expressed at lower levels in florets of all three white cultivars compare with the green cultivars, while *CmCOL16a* and *CmERF* showed lower expression in florets of some white cultivars compared to the green cultivars (Figure 10C). Thus, these four genes are key TFs regulating the green flower color in chrysanthemum.

## 4. Discussion

Flower color in chrysanthemum is an important ornamental trait. Green chrysanthemums have been generated through breeding but are difficult to obtain owing to the absence of green flowers in the same genus [1]. New cultivars with green petals are desired to satisfy consumers, but the molecular mechanisms underlying petal-specific chlorophyll accumulation that results in green chrysanthemum flowers are not fully understood. In previous studies on green flowers, several commercial cultivars or mutant lines were used for transcriptome analyses [7,14]. In this study, we performed a transcriptome analysis using pooled samples from a segregating F_1_ population. The parental cultivars *Chrysanthemum × morifolium* “Lüdingdang” and *C. vistitum* differ greatly in terms of the color of the ray florets and other traits. For our analyses, we pooled the extreme green and white individuals from their hybrid progeny to eliminate potential interference from these other traits.

Segregating populations are mostly used for forward genetics—e.g., genetic analysis [25,35,36] and quantitative trait loci mapping [37,38]. Mixed samples from a segregating population have been used for bulked segregant analysis coupled with whole-genome sequencing, bulked segregant RNA-seq [39,40], and transcriptome analysis of human diseases [41]; however, they have rarely been used in studies on green chrysanthemum. By analyzing a segregating F_1_ population, we expected genes related to floret color to be highly enriched. We examined three different stages of flower development in order to identify key genes related to green flower color (i.e., those showing the greatest difference in expression between green and white pools) and eliminate the influence of other genes related to flower development. Some of the candidate genes were similar to those identified in previous studies [14], demonstrating that in the absence of bud mutants, it is possible to establish extreme phenotype pools from a segregating population for transcriptome sequencing and analysis.

The transcriptome analysis identified 113 genes involved in chlorophyll metabolism, of which 24 were DEGs with higher expression in green pools than in white pools at all three examined stages of flower development. Previous studies on green flowers of chrysanthemum and carnation showed that the lower chlorophyll content of non-green carnation petals as compared to pale green petals was due to differences in chlorophyll metabolism [7], a low rate of biosynthesis, and a high rate of degradation resulting in the absence of chlorophyll in white chrysanthemum petals [14]. A similar finding was reported in the analysis of the process of lily petals changing from green to white [13]. In contrast, we found that both the biosynthesis and degradation of chlorophyll were reduced in green pools compared to white pools. We speculate that this reflects a global suppression of chlorophyll metabolism in white ray florets.

Previous transcriptome analyses of green flowers mainly focused on chlorophyll metabolism, while less attention has been given to other photosynthesis-related pathways. By WGCNA and KEGG pathway enrichment analyses, we determined that the green color of chrysanthemum ray florets is influenced by both chlorophyll metabolism and photosynthesis, as indicated by the enrichment of the “Photosynthesis”, “Carbon fixation in photosynthetic organisms”, “Photosynthesis-antenna proteins”, “Carbon metabolism”, and “Glyoxylate and dicarboxylate metabolism” KEGG pathways. Similar results were reported for the chrysanthemum variety “Anastasia Dark Green” and its white-flowered bud sport, in which genes related to the photosystem, thylakoid, plastid, and chloroplast were differentially expressed between samples with green and white ray florets [42].

The TEM analysis revealed that the rate of chloroplast disintegration was much slower in green petals than in white petals during chrysanthemum flower development. This is consistent with previous observations of the thylakoid membrane in chrysanthemum with white ray florets [14] and the finding that chloroplasts were larger and the grana thylakoid membrane structure was more clearly defined in chrysanthemum disc florets with a green corolla as compared to a yellow corolla [9]. Under heat stress, the petals of *N. tabacum* and *A. thaliana* flowers overexpressing SOC1 and SOC1-like genes produced normal chloroplasts and had a green color, whereas flowers of wild-type plants had non-photosynthetic plastids [18]. These findings suggest that the presence of a normal chloroplast in petals is a key factor affecting petal color.

The black module of the WGCNA comprised 27 TFs belonging to 19 TF families. Although they were differentially expressed between green and white pools, their expression patterns during the three stages of flower development were similar, suggesting that they were not directly involved in flower development. The key TFs in this module were CmCOLa, CmCOLb, and CmERF. Additionally, CmbHLH was identified in the expression analysis; although it was not part of the same co-expression network as the black module from the WGCNA, its expression level varied with flower color across samples (Figure 10C). CmCOLa and CmCOLb have a B-box domain within the N-terminal zinc finger and a C-terminal CCT domain belonging to the CONSTANS-like family [43]. The *COL16* gene, which was identified in a transcriptome analysis of the chrysanthemum (*C. morifolium* Ramat.) cultivar “Feeling Green” and its bud sports [14], has a nucleotide sequence that is 97% similar to that of *CmCOLb*. In another study, transgenic petunia plants overexpressing *PhCOL16a* had pale green corollas with a higher chlorophyll content than the wild type as well as higher chlorophyll synthesis activity and more robust chloroplast structure [44]. In our study, *CmCOLa* had higher betweenness and degree than *CmCOLb* in the gene co-expression network, suggesting that it has a more prominent role in the determination of green flower color. CmERF belonging to the AP2/ERF TF family regulates primary and secondary metabolism, growth and development, and responses to environmental stimuli [45,46]. The expression of some ERFs is negatively correlated with chlorophyll accumulation in fruit and leaves. In a study of citrus fruit (*Citrus reticulata* Blanco cv. Ponkan), CitERF6 and CitERF13 expression was enhanced by the hormone ethylene, leading to chlorophyll degradation and fruit de-greening [47]. In yellow leafy head of *Brassica rapa* (subsp. *pekinensis*), some chlorophyll synthesis-related genes were found to be downregulated while some ERF TFs were upregulated [48]. An ERF TF was previously identified in chrysanthemum that was highly expressed in green-flowered cultivars but had low expression in white-flowered cultivars [14]. Consistent with this report, in our study, there were three ERFs identified as hub genes whose expression was positively correlated with chlorophyll content in green-flowered chrysanthemum cultivars. Thus, TFs in the ERF family have different regulatory mechanisms for chlorophyll metabolism in different species and different tissues. CmbHLH belongs to the bHLH family, which is the second largest TF superfamily [49] and is involved in the regulation of many developmental and metabolic processes [50]. In lily, 25 TF gene families that were co-expressed with chlorophyll metabolism-related genes were identified by WGCNA, with the bHLH family being the most highly represented [13]. We also found two genes encoding C2H2 family TFs whose expression was positively correlated with chlorophyll content. C2H2 zinc finger proteins perform multiple functions in plants [51], participating in growth and development (including of flowers) and the response to abiotic stress [52], although a role in the regulation of chlorophyll metabolism has not been reported.

In summary, in this study, we performed a transcriptome analysis and a WGCNA using mixed samples from a segregating F_1_ population of chrysanthemums with white and green florets and identified genes related to photosynthesis and chlorophyll metabolism, including the TF-encoding genes *CmCOL16a*, *CmCOL16a*, *CmERF*, and *CmbHLH*, that are responsible for the green color of chrysanthemum flowers. Our results demonstrate that it is possible to perform a transcriptome analysis using a mixed population of individuals with extreme phenotypes when an appropriate mutant is lacking. Moreover, they provide a basis for investigating the production of green flowers in other plant species and can guide future efforts to improve the color palette of chrysanthemum flowers through genetic engineering.

## Figures and Tables

**Figure 1 genes-12-00449-f001:**
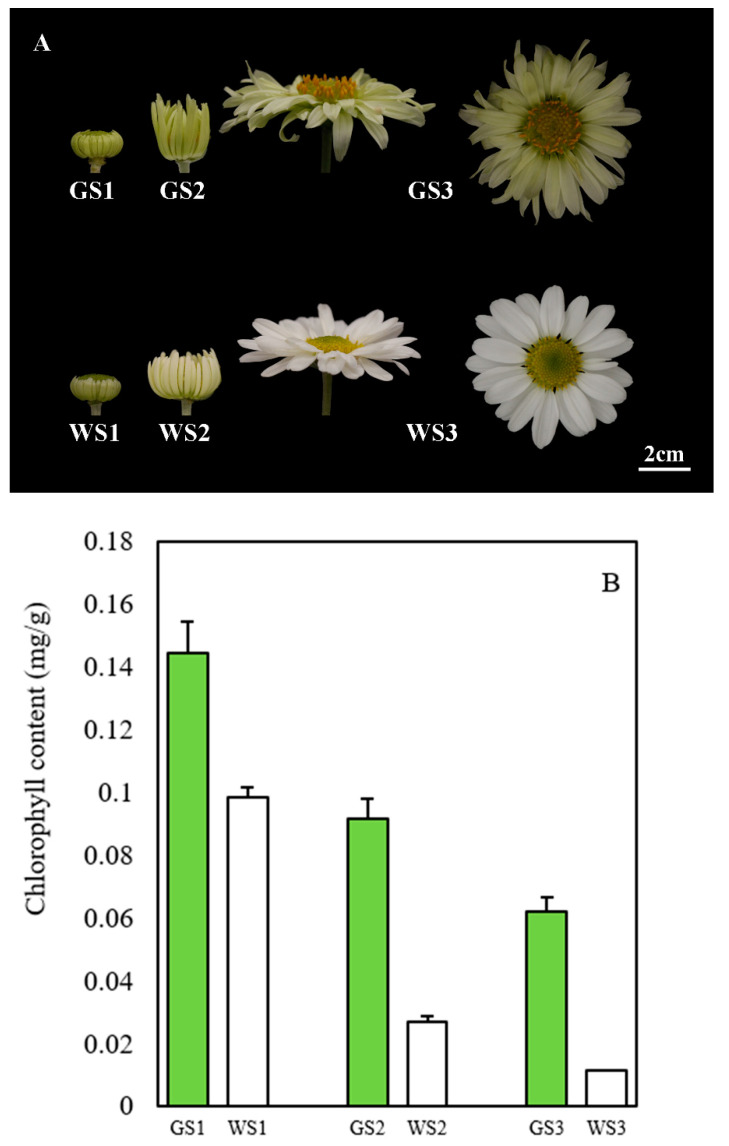
Phenotype and chlorophyll accumulation at three stages of flower development. (**A**) Floret color at each stage of flower development. (**B**) Chlorophyll content of petals from green and white pools at each stage.

**Figure 2 genes-12-00449-f002:**
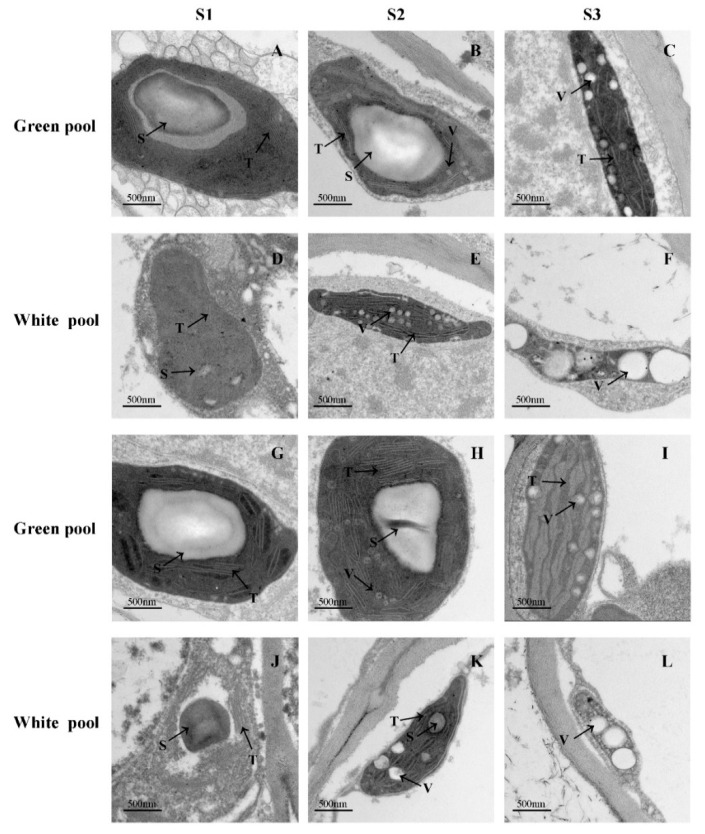
TEM images of plastid ultrastructure. (**A**–**C**,**G**–**I**) Plastid ultrastructure in the epidermal cells (**A**–**F**) and mesophyll cells (**G**–**L**) of ray florets of green (**A**–**C**,**G**–**I**) and white (**D**–**F**,**J**–**L**) pools. S, starch granule; T, thylakoids; V, lipid vesicles.

**Figure 3 genes-12-00449-f003:**
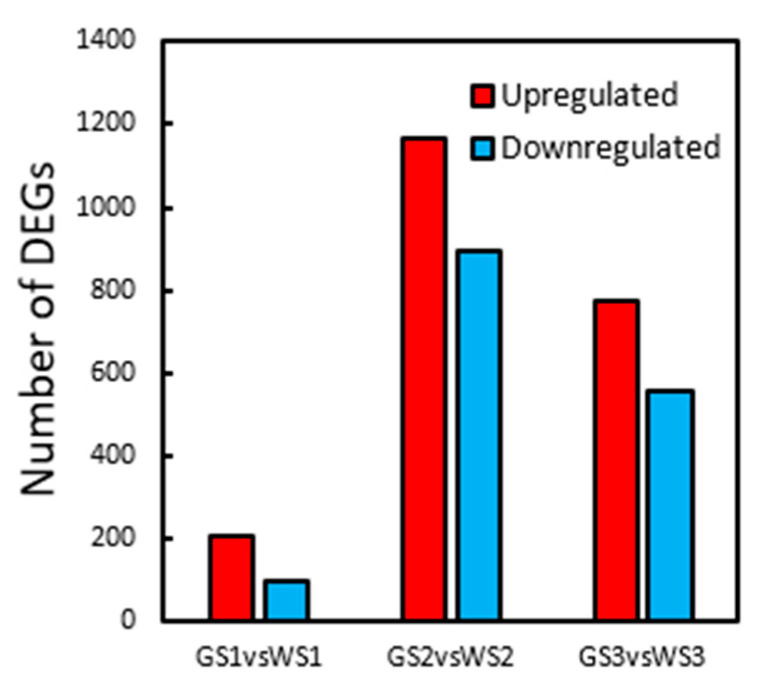
Differentially expressed genes (DEGs) between green and white pools at each stage of flower development.

**Figure 4 genes-12-00449-f004:**
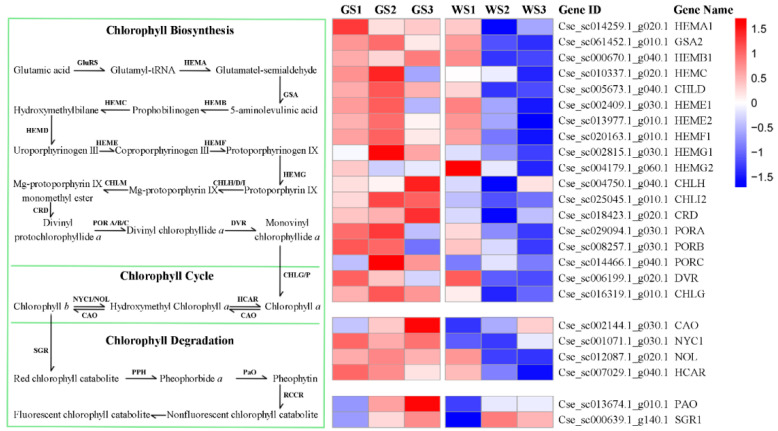
Expression heatmap of DEGs involved in chlorophyll metabolism in different samples.

**Figure 5 genes-12-00449-f005:**
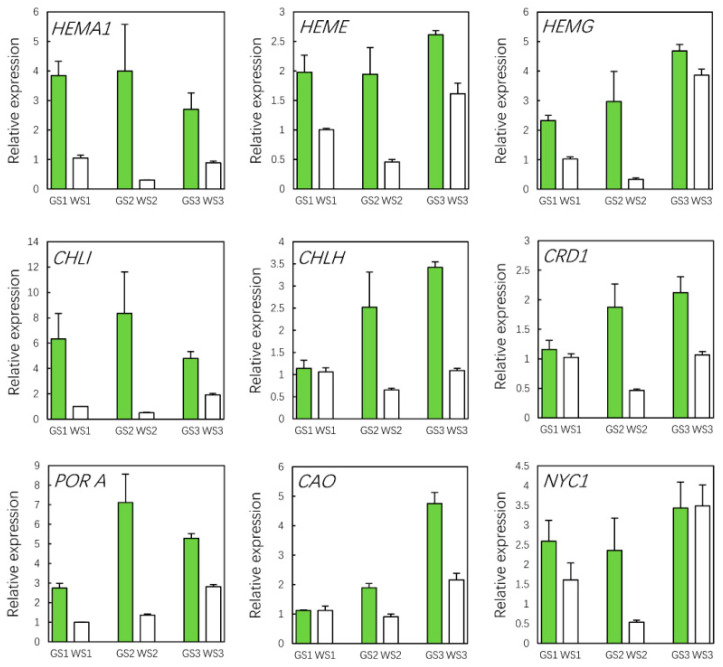
qRT-PCR analysis of DEGs involved in chlorophyll metabolism in different samples.

**Figure 6 genes-12-00449-f006:**
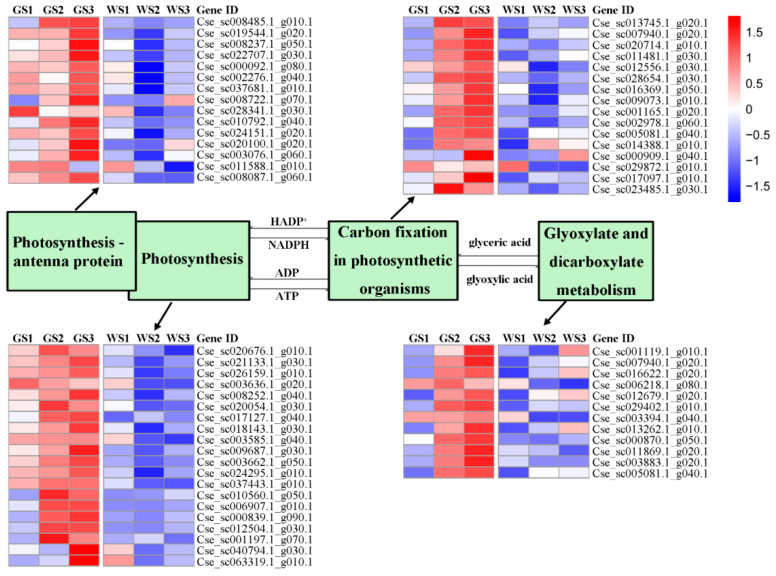
Expression heatmap of genes in key pathways that are differentially expressed between green and white pools at three stages of flower development.

**Figure 7 genes-12-00449-f007:**
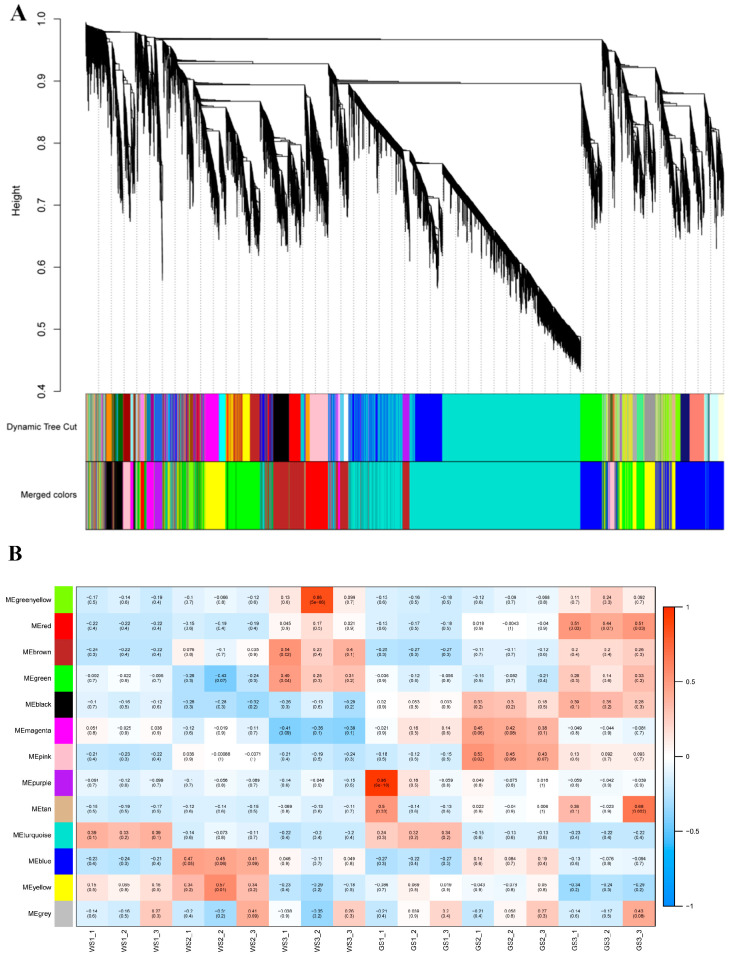
Co-expression modules determined by weighted gene co-expression network analysis (WGCNA). (**A**) Cluster dendrogram of genes in the WGCNA. (**B**) Heatmap of correlations between module eigengenes and samples. Genes in the black module showed the highest positive correlation with flower color (i.e., chlorophyll content).

**Figure 8 genes-12-00449-f008:**
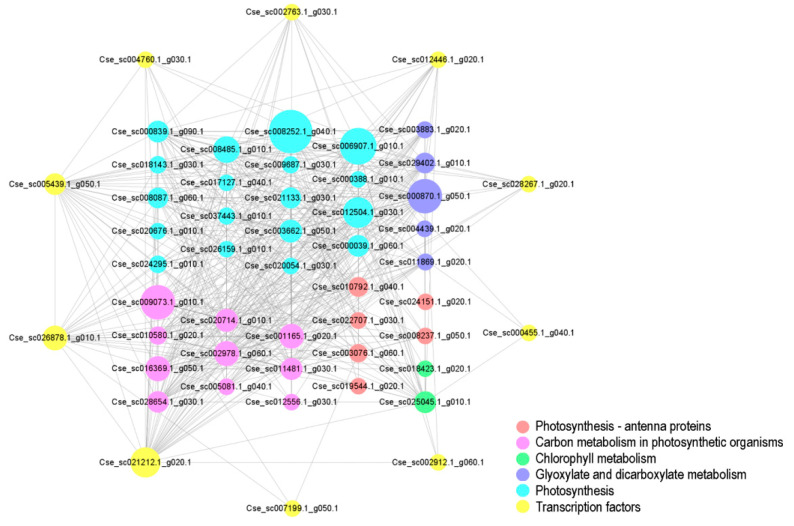
Network of genes in the black module. The network was constructed from 51 genes based on edge weight values. The size of the nodes represents the betweenness centrality of the node, with larger nodes having more connections with other nodes.

**Figure 9 genes-12-00449-f009:**
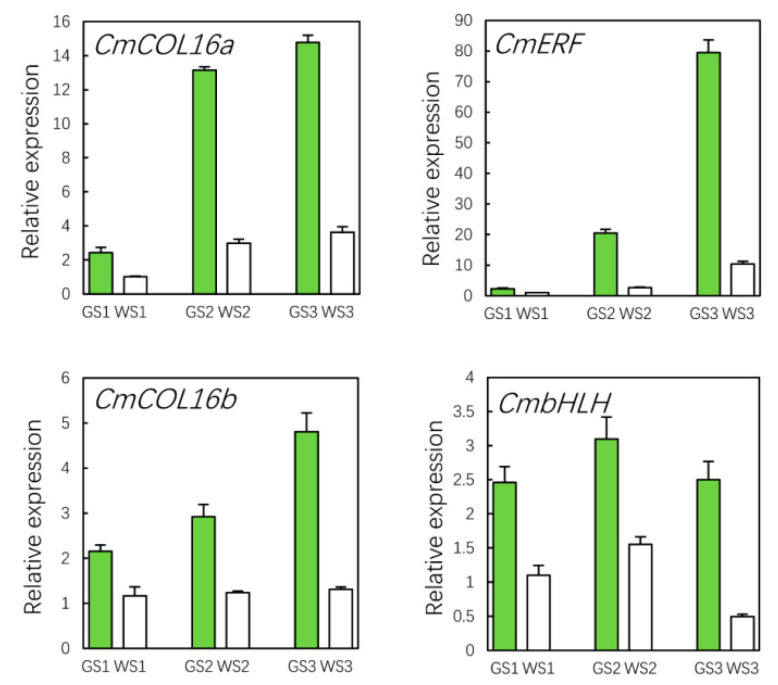
qRT-PCR analysis of hub genes in green and white pools at different stages of flower development.

**Figure 10 genes-12-00449-f010:**
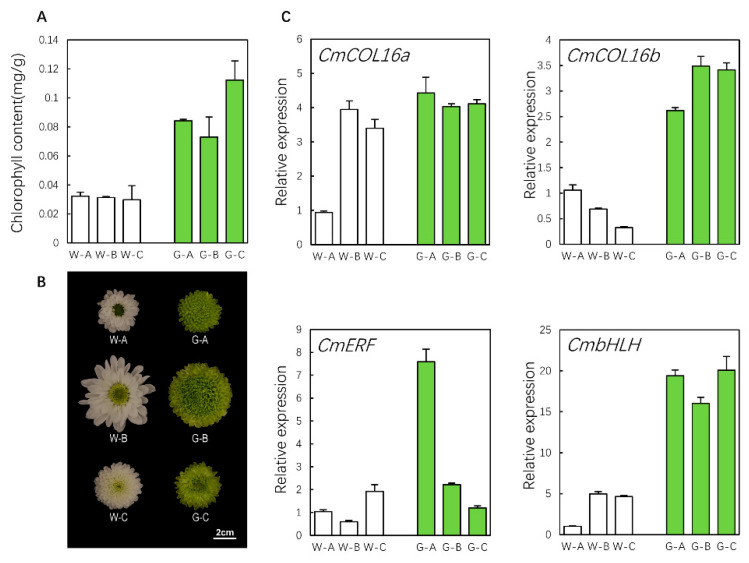
qRT-PCR analysis of hub genes in petals of green and white ray florets of chrysanthemum. (**A**–**C**) Chlorophyll content (**A**), color (**B**), and expression of hub genes in the six commercial cultivars (**C**).

**Table 1 genes-12-00449-t001:** Summary of transcriptome sequencing data and transcriptome assembly.

Samples	Library	Raw Reads	Clean Reads	Clean Bases	Error (%)	Q20 (%)	Q30 (%)	GC (%)
GS1_1	FRAS192310180-1r	69,909,924	67,974,534	10.2G	0.02	98.22	94.50	41.89
GS1_2	FRAS192310181-1r	67,154,326	65,518,128	9.83G	0.02	98.28	94.64	42.19
GS1_3	FRAS192310182-1r	79,385,102	77,582,694	11.64G	0.02	98.16	94.41	42.38
WS1_1	FRAS192310183-1r	78,199,758	76,158,198	11.42G	0.02	98.26	94.69	42.45
WS1_2	FRAS192310184-1r	68,968,074	67,390,906	10.11G	0.02	98.25	94.55	42.33
WS1_3	FRAS192310185-1r	71,120,720	69,206,430	10.38G	0.02	98.10	94.24	42.40
GS2_1	FRAS192310186-1r	74,730,896	72,493,234	10.87G	0.02	98.19	94.48	42.38
GS2_2	FRAS192310187-1r	72,994,076	70,710,496	10.61G	0.02	98.25	94.63	42.26
GS2_3	FRAS192310188-1r	73,085,728	71,432,126	10.71G	0.02	98.11	94.23	42.16
WS2_1	FRAS192310189-1r	68,345,294	66,024,344	9.9G	0.02	98.16	94.45	41.92
WS2_2	FRAS192310190-1r	81,521,172	79,358,278	11.9G	0.02	98.19	94.56	41.69
WS2_3	FRAS192310191-1r	69,440,584	67,776,290	10.17G	0.02	98.08	94.24	41.79
GS3_1	FRAS192042324-1r	76,123,912	74,489,046	11.17G	0.03	97.90	93.72	41.99
GS3_2	FRAS192042325-1r	76,239,258	74,539,470	11.18G	0.03	98.00	93.99	42.07
GS3_3	FRAS192042326-1r	75,168,204	73,433,218	11.01G	0.03	97.92	93.74	42.05
WS3_1	FRAS192042327-1r	78,303,734	76,184,148	11.43G	0.03	97.84	93.61	41.70
WS3_2	FRAS192042328-1r	78,068,928	76,158,448	11.42G	0.03	97.92	93.77	41.68
WS3_3	FRAS192042329-1r	79,621,212	77,866,790	11.68G	0.03	97.82	93.55	41.99

**Table 2 genes-12-00449-t002:** Enrichment of key KEGG pathways among DEGs.

KEGG Pathway	No. in All Unigenes	No. in DEGs
Photosynthesis	151	20
Photosynthesis—antenna proteins	65	15
Chlorophyll metabolism	113	24
Carbon fixation in photosynthetic organisms	230	16
Glyoxylate and dicarboxylate metabolism	204	12

DEG, differentially expressed gene; KEGG, Kyoto Encyclopedia of Genes and Genomes.

**Table 3 genes-12-00449-t003:** Distribution of genes related to key pathways in 26 modules.

Module		Number of DEGs Related to Key Processes in Chrysanthemum				
	Total DEGs	Photosynthesis—Antenna Proteins	Photosynthesis	Carbon Fixation in Photosynthetic Organisms	Glyoxylate and Dicarboxylate Metabolism	Chlorophyll Metabolism
Red	116	5	0	0	0	0
Black	97	6	20	10	7	5
Other	1621	0	0	0	0	0

DEG, differentially expressed gene.

## Data Availability

Data supporting the reported results can be found at the National Center for Biotechnology Information (https://www.ncbi.nlm.nih.gov/bioproject/) with the BioProject ID PRJNA704494.

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
