# Peer review of "Identification of Chlorophyll Metabolism- and Photosynthesis-Related Genes Regulating Green Flower Color in Chrysanthemum by Integrative Transcriptome and Weighted Correlation Network Analyses"

_genes, 2021, doi:10.3390/genes12030449_

Round 1

Reviewer 1 Report

This paper has academic significance as a research paper on the genes involved in regulating green flower color in chrysanthemum using WGCNA.
These data was interesting and helpful to understand the mechanism of colors of chrysanthemums. But I have some questions in this research.

Line113-114: Add information about the chrysanthemum varieties purchased from the market. (W-A, G-A, W-B, G-B....etc)
Line 264-270/349: As far as I know, chrysanthemum is a polyploid. what is the status of gene duplication in TFs and chloroplast related genes?, Is there only one TFs or chloroplast genes?
Line 296: The resolution of Figure 7B is not good. Please replace the picture as the content cannot be checked.
Line 445-455: This statement was inaccurate according to the data in this paper
The shapes of the three samples look similar (Figure 10B: G-A, G-B, G-C). Do you think this has nothing to do with gene expression.....TFs?

Author Response

Response to Reviewer 1 Comments

Point 1: Add information about the chrysanthemum varieties purchased from the market. (W-A, G-A, W-B, G-B....etc)

Response 1: In our study, we focused on the ray florets of chrysanthemum, so we added the shape information of the ray florets of the six chrysanthemum varieties purchased from the market in addition to the color at Line113-117.

Point 2 : As far as I know, chrysanthemum is a polyploid. what is the status of gene duplication in TFs and chloroplast related genes?, Is there only one TFs or chloroplast genes?

Response 2: Thank you, Chrysanthemum morifolium is considered to be hexaploidy. There are homologous genes in the genome, different homologous genes may have different expression levels. In our study, in the comparison with NCBI of the 4 TFs that we identified in the research, homologous genes with SNP were not found. And the study on citrus (Citrus. junos cv. Ziyang xiangcheng) showed that only 0.8% of detected genes are found significantly differentially expressed between 2x and 4x leaves of citrus(Tan et al. 2015). So we considered that the polyploidy of chrysanthemums will not affect our conclusions.

But the formulation ‘No. in all unigenes’ in Table 2 is inaccurate. so, we revise ‘genes’ to ‘unigenes’

(Tan, Feng-Quan; Tu, Hong; Liang, Wu-Jun; Long, Jian-Mei; Wu, Xiao-Meng; Zhang, Hong-Yan; Guo, Wen-Wu (2015). Comparative metabolic and transcriptional analysis of a doubled diploid and its diploid citrus rootstock (C. junos cv. Ziyang xiangcheng) suggests its potential value for stress resistance improvement. BMC Plant Biology, 15(1), 89–. doi:10.1186/s12870-015-0450-4)

Point 3: The resolution of Figure 7B is not good. Please replace the picture as the content cannot be checked.

Response 3: Thank you, and images with high resolution have been replaced.

Point 4: This statement was inaccurate according to the data in this paper

Response 4: The original description is not appropriate, I have made the adjustments at Line457-460,

Point 5: The shapes of the three samples look similar (Figure 10B: G-A, G-B, G-C). Do you think this has nothing to do with gene expression.....TFs?

Response 5: Thank you, the shape of that three samples look similar, but in this study, the 4 TFs which were verified by these samples were co-expressed with the chlorophyll metabolism and photosynthesis related metabolic pathways, it is speculated that the high expression of these transcription factors in green chrysanthemum can promote the synthesis of chlorophyll and lead to different flower colors, whether they also affects the flower type, it needs further study. 

Kind regards,
Sincerely yours,
Jing Luo & CaiYun Wang           

Reviewer 2 Report

The authors (Fu et al) analyzed chlorophyll metabolism and photosynthesis- related genes in green-flowered chrysanthemums. Authors used RNA-Seq. analysis and compared the expression levels between green-flowers and white-flowers. They showed that some photosynthesis and chlorophyll- metabolism related genes were up- or down-regulated. The results present newly information about these two synthesis pathway. Moreover, they demonstrated that four transcription factors (CmCOL16a, CmCOL16b, CmERF and CmbHLH) strongly expressed in green flowers (ray florets).

I think that the author’s opinion is very interesting and the results are reasonable. The focus of the paper is well phrased.

In my opinion, this manuscript is acceptable for publication on Genes.

Author Response

Response to Reviewer 2 Comments

Thank you for your review. According to your Comments and another reviewer, some English language and style have been corrected and optimized in the manuscript, the details can be found in the cover letter.

Kind regards,
Sincerely yours,
Jing Luo & CaiYun Wang 
